# Interplay of surface interaction and magnetic torque in single-cell motion of magnetotactic bacteria in microfluidic confinement

Agnese Codutti[1,2,3†], Mohammad A Charsooghi[1,4†], Elisa Cerdá-Doñate[1†], Hubert M Taïeb[1], Tom Robinson[2]*, Damien Faivre[1,5]*, Stefan Klumpp[2,6]*

[1]Biomaterials department, Max Planck Institute of Colloids and Interfaces, Potsdam, Germany; [2]Theory and Bio-systems department, Max Planck Institute of Colloids and Interfaces, Potsdam, Germany; [3]Biological physics and Morphogenesis group, Max Planck Institute of Dynamics and Self-Organization, Göttingen, Germany; [4]Department of Physics, Institute for Advanced Studies in Basic Sciences (IASBS), Zanjan, Islamic Republic of Iran; [5]Aix Marseille Université, CNRS, CEA, BIAM, Saint Paul lez Durance, France; [6]Institute for the Dynamics of Complex Systems, University of Göttingen, Göttingen, Germany

*For correspondence:
tom.robinson@mpikg.mpg.de
(TR);
damien.faivre@cea.fr (DF);
stefan.klumpp@phys.uni-goettingen.de (SK)

†These authors contributed equally to this work

Competing interest: The authors declare that no competing interests exist.

**Abstract** Swimming microorganisms often experience complex environments in their natural habitat. The same is true for microswimmers in envisioned biomedical applications. The simple aqueous conditions typically studied in the lab differ strongly from those found in these environments and often exclude the effects of small volume confinement or the influence that external fields have on their motion. In this work, we investigate magnetically steerable microswimmers, specifically magnetotactic bacteria, in strong spatial confinement and under the influence of an external magnetic field. We trap single cells in micrometer-sized microfluidic chambers and track and analyze their motion, which shows a variety of different trajectories, depending on the chamber size and the strength of the magnetic field. Combining these experimental observations with simulations using a variant of an active Brownian particle model, we explain the variety of trajectories by the interplay between the wall interactions and the magnetic torque. We also analyze the pronounced cell-to-cell heterogeneity, which makes single-cell tracking essential for an understanding of the motility patterns. In this way, our work establishes a basis for the analysis and prediction of microswimmer motility in more complex environments.

## Editor's evaluation

The manuscript reports results of a combined experimental and numerical investigation of magnetotactic bacteria in strong spatial confinement and under the influence of an external magnetic field. Single cells are trapped in micrometer-sized microfluidic chambers, and a variety of trajectories are found, which depend on the chamber size and the strength of the magnetic field. A detailed understanding of swimming in simple controlled confinement is essential to predict the behavior of motile microorganisms in the complex environments of their natural habitat.

## Introduction

Microswimmers and other self-propelled microscopic particles have received extensive attention in recent years, both directed toward the fundamental physical principles of their propulsion mechanisms and their rich collective behavior (*Bechinger et al., 2016*; *Ezhilan et al., 2015*; *Lauga and Powers, 2009*), and toward their potential for environmental and biomedical applications including as drug carriers, infertility cures, bioremediation, or MRI imaging (*Bente et al., 2018*; *Felfoul et al., 2016a*; *Felfoul et al., 2016b*; *Ford and Harvey, 2007*; *Magdanz and Schmidt, 2014*; *Peyer et al., 2013b*; *Pieper and Reineke, 2000*; *Schwarz et al., 2017*; *Sherwood et al., 2003*; *Timmis and Pieper, 1999*). Envisioned applications of microswimmers often imply their use in complex environments that may present a variety of obstacles, confinement to pores, fluid flows, complex fluids, or other complications (*Bente et al., 2018*; *Felfoul et al., 2016b*) compared to the motion in simple aqueous solutions that is typical under lab conditions (*Bennet et al., 2014*; *Lefèvre et al., 2014*; *Popp et al., 2014*). The same is true for the environments that form the natural habitats of microorganisms that are studied as biological or hybrid microswimmers, and include sediments, soils, as well as the natural habitats within the bodies of animals and humans (*Dehkharghani, 2017*; *Dillon et al., 1995*; *Duffy et al., 1995*; *Ford and Harvey, 2007*; *Raatz et al., 2015*; *Reichhardt and Olson Reichhardt, 2014*; *Theves et al., 2015*; *Torkzaban et al., 2008*). Several recent studies have therefore started to address their motion in complex environments such as model porous systems and confined spaces using microfluidic devices; for example using arrays of pillars (*Aleklett et al., 2018*; *Anbari et al., 2018*), ratchet-like sorting devices (*Galajda et al., 2007*; *Reichhardt and Reichhardt, 2017*; *Wan et al., 2008*), or droplets (*Vincenti et al., 2019*).

One situation of particular interest is the confinement of microswimmers to a small space. In this case, the steric and hydrodynamic interactions with the confining walls have a strong impact on the motility and swimmer trajectories (*Theves et al., 2015*). Several types of microswimmers have been shown to accumulate near walls and to be guided by walls in microchannels with complex geometries (*Berke et al., 2008*; *Denissenko et al., 2012*; *Kantsler et al., 2013*). These interactions have been discussed theoretically based on hydrodynamics as well as steric interactions of the swimmers' bodies and flagella with the walls (*Fily et al., 2014*; *Rode et al., 2019*; *Spagnolie et al., 2015*; *Wysocki et al., 2015*). Moreover, some microorganisms show behavioral responses to collision with surfaces, for example by changing their mode of self-propulsion (*Kühn et al., 2017*). The effect of confinement on their behavior can be studied using microfluidic devices that trap microswimmers and confine them in three dimensions (3D) for extended observation of individual swimmers, which allow us to study the effect of various parameters such as changes in magnetic field on the same bacterium. However, few such studies have been reported so far. One notable exception is a recent study of the alga *Chlamydomonas reinhardtii*, which was observed to swim along the wall and to accumulate in regions with a large wall curvature (*Ostapenko et al., 2018*). However, this report only focused on a dumbbell shape model to represent the wall steric interaction, which is not always applicable to different swimmers due to the variety of cell shapes and swimming patterns (pusher or puller type for example).

The tactic behavior of microorganisms is known to help their navigation and proliferation in complex environments. Accordingly, a line of research has also developed, which seeks to understand the directionality of microswimmers and how to control them due to their tactic behaviors or due to interactions with external fields (*Codutti et al., 2019*; *Ford and Harvey, 2007*; *Hassan et al., 2016*). This regard, magnetic fields are particularly interesting, as they allow the remote steering of microswimmers exhibiting a magnetic moment, a feature suitable for the remote control of the swimmers performing biomedical tasks in the human body (*Bente et al., 2018*). Such microswimmers include magnetotactic bacteria (*Bente et al., 2019*; *Frankel, 1984*; *Frankel et al., 2006*), that is bacteria equipped with a magnetic moment due to dedicated organelles, the magnetosomes, as well as biohybrid and synthetic magnetic swimmers (*Bente et al., 2018*; *Huang and Mei, 2015*; *Park et al., 2017*; *Peyer et al., 2013a*; *Stanton et al., 2017*; *Vach et al., 2013*). The use of magnetic microswimmers in small confined spaces such as microfluidic traps opens the possibility to study the interplay of confinement and directionality imposed by the external magnetic field. An understanding of this interplay is crucial to understand the navigation of biological microswimmers in complex environments as well as to develop strategies for steering microswimmers through such environments.

Here, we therefore investigate the swimming of the magnetotactic bacterium *Magnetospirillum gryphiswaldense* (MSR-1) confined to circular traps, both in the presence and absence of an external

magnetic field that guides the swimming directionality. To that end, we designed a microfluidic trapping platform, featuring actuatable elastomeric PDMS membranes to create defined micrometer-sized containers. This approach allows us to observe and track individual bacteria for extended periods of time and to account for considerable heterogeneities in their physical properties and swimming behaviors. Single-cell analysis allows us to pinpoint the main characteristics determining the trajectories, which would otherwise be lost and averaged out in a cell population approach. We complement our experimental investigation with simulations using a variant of an active Brownian particle (ABP) model, providing an effective general model to describe the complex wall-bacterium interaction. Combining the experimental results with the simulations allows us to explain the variety of observed behaviors by the interplay of two torques, induced on the one hand by the magnetic field and on the other hand by the interaction with the confining walls. Overall, our approach combining microfluidic trapping, observation of individual swimmers, and interpretation of the observations with the help of modeling provides a path toward understanding and predicting swimmers in other, more complex environments.

## Results

### Microfluidic trapping of single bacteria

Confining individual bacteria in a small, closed volume provides an opportunity to study the motility of individual bacteria by long-time observation of the same cell. Microfluidics offers the possibility of microfabricating such small features on a length scale similar to that of a single cell, thus emphasizing their interactions with the confining walls. However, constructing a well-defined and reproducible volume that is sealable is challenging. Previous approaches such as the so-called 'Quake's valves' (*Thorsen et al., 2002*) or donut-shaped valves (*Robinson et al., 2013*) are not applicable as they result in edges where bacteria can become permanently trapped, or generate volumes well beyond a single-cell size, respectively. Therefore, we developed a microfluidic trapping system based on a pressure actuated ceiling (*Figure 1* and *Figure 1—figure supplements 1 and 2*) to trap and study individual magnetotactic bacteria. In this multilayer microfluidic strategy, a bacterial solution is placed inside a microfluidic channel whose ceiling (a PDMS membrane) is patterned with open cylinders (*Figure 1—figure supplement 2c*). Due to the elastic properties of thin PDMS membranes, when a downward force is applied on the patterned ceiling, it gets displaced toward the bottom of the channel. This results in the cylinders becoming sealed (i.e. traps), resulting in the confinement of bacteria inside them (*Figure 1A and B*). The circular design was fabricated directly into the channel ceiling for size reproducibility and control, thus ensuring a 2D environment where we can tune the trap dimensions to influence the swimming behavior (i.e. diameter).

Single bacteria were isolated in circular microfluidic traps of different dimensions with radii of 7.5, 12.5, 17.5, 25, and 45 µm (for comparison, the length of the bacterial cell body is typically 3 µm). The trap size is reproducible up to a variability in the radii of 1–2 µm (see 'Materials and methods – Microfluidic trap characterization'). The height of the trap is fixed at 10 µm, limiting the motion to the x-y plane and providing simple, quasi-2D trajectories. Successful confinement of individual bacteria in these microfluidic traps allowed us to image the same cell for up to 1 hr. *Figure 1c* shows the histogram of the number of trapped bacteria for different cell concentrations (given by the optical density $OD_{565}$). In this work, we aimed at characterizing the motion of individual bacteria, therefore the cell concentration was adjusted (OD = 0.03–0.05) such that typically only one bacterium is trapped in the smaller microfluidic traps (from 7.5 to 17.5 µm radius), the rare traps containing more than one bacterium were excluded from the analysis. For the largest trap size (45 µm), it was not possible to trap single bacteria, so we analyzed cases with two or three bacteria in one trap (*Figure 1—figure supplement 3*), for the traps of 25 µm, we analyzed traps containing one to three bacteria. To test for potential effects of interactions between the bacteria, we compared trajectories in the latter traps containing different numbers of bacteria and found no systematic dependence on that number (see below). *Figure 1d* shows a typical trajectory of a bacterium in the microfluidic trap of 45 µm in the absence of any magnetic field (ambient magnetic field is cancelled out, see 'Materials and methods – Microscopy and image stabilization'; trajectories in smaller microfluidic traps are shown in *Figure 1—figure supplement 4*, a representative movie in *Video 1*). Because the bacterium swims very close to the wall, an image stabilization and a background subtraction processing was implemented to track

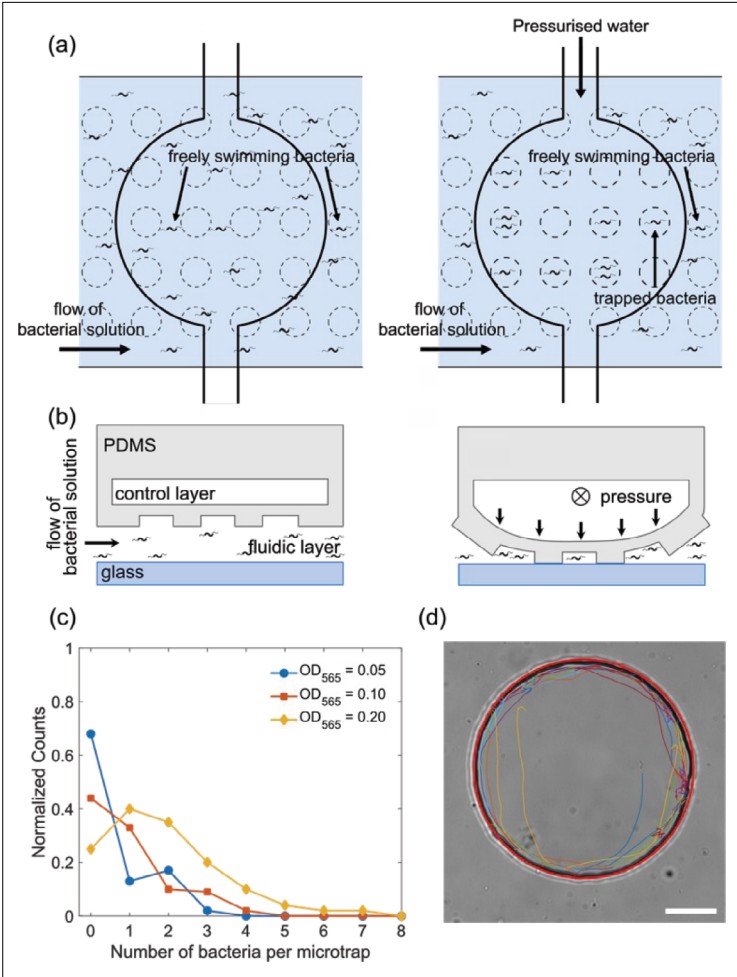

**Figure 1.** Trapping and tracking of bacteria in microfluidic traps. (**a**) Schematic top view of how the microfluidic trap system achieves confinement of bacteria using multiple layers: a fluidic layer (bottom layer) and a control layer (top layer). Free bacteria are introduced using the lower fluidic layer and by applying pressure to the top control layer the valve deforms and the bacteria are confined in the micron-sized traps of a defined diameter. (**b**) Schematic side view of the valve in panel (**a**), see Materials and methods 'Fabrication of the master molds and microfluidic systems' and 'Operation of the chip' sections for more details. (**c**) Histogram of the number of bacteria isolated in one microfluidic trap of radius 12.5 µm at three different bacteria concentrations of $OD_{565}$ 0.05, 0.1, and 0.2. (**d**) Bright-field microscopy image of a typical 45 µm radius trap with the extracted trajectory of the bacteria (different segments of the same trajectory are depicted with colored lines). A red circle is fitted to visualize the microfluidic trap perimeter. Scale bar is 20 µm.

The online version of this article includes the following figure supplement(s) for figure 1:

**Figure supplement 1.** Schematics of the device configuration.

**Figure supplement 2.** Microfluidic chip.

**Figure supplement 3.** Number of bacteria in the microtraps.

**Figure supplement 4.** Bacteria trajectories in traps of different sizes.

the motion (see 'Materials and methods – Microscopy and image stabilization'), but even with these improvements the tracking algorithm cannot track the bacterium continuously. Therefore, continuous fragments (43±16 fragments) of trajectories are shown in different colors in *Figure 1d*.

## Motion in confinement

For all microfluidic trap dimensions and in the absence of applied magnetic fields, we observed that bacteria predominantly swim along the walls and only occasionally move to the interior of the trap (*Figure 1d*). To quantify this behavior, we determined the radial probability distribution functions of

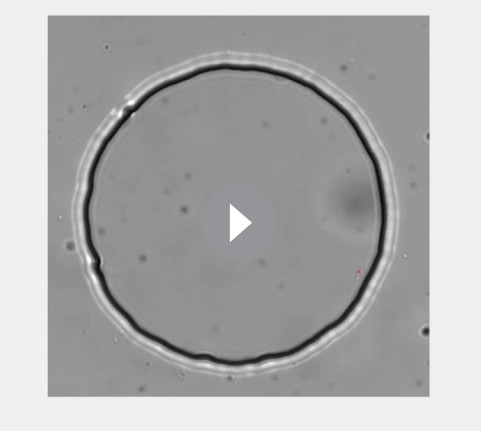

**Video 1.** Real-time magnetotactic bacteria (MSR-1) swimming in a microtrap with radius of 45 μm in the absence of external magnetic field. Imaged with 40× objective lens (NA = 0.6, air, Nikon) and a sCMOS camera (2560×2,160 pixels; Zyla, Andor Technology) with the rate of 50 fps.

https://elifesciences.org/articles/71527/figures#video1

the trap occupancy (*Figure 2*). Swimming along the walls is reflected in a pronounced peak in that distribution, situated within few micrometers from the border of the microfluidic trap (see *Table 1* for the peak position values), and with a broad tail that extends into the center. The observed behavior is similar to the one reported for algae in confinement (*Ostapenko et al., 2018*), even though the bacterium is hydrodynamically different from an alga, since it possesses a spiral shape body (*Frankel, 1984*; *Frankel et al., 2006*) instead of a spherical one (*Ostapenko et al., 2018*); two rotating flagella at each pole of the body (*Frankel, 1984*; *Frankel et al., 2006*) instead of two beating flagella in the front (*Ostapenko et al., 2018*); and, most probably, displays a pusher and puller dynamic like other biflagellate magnetotactic bacteria (*Murat et al., 2015*) instead of breaststroke swimming (*Ostapenko et al., 2018*).

Occasionally, bacteria are seen to reverse their motion. These events are rare and are observed

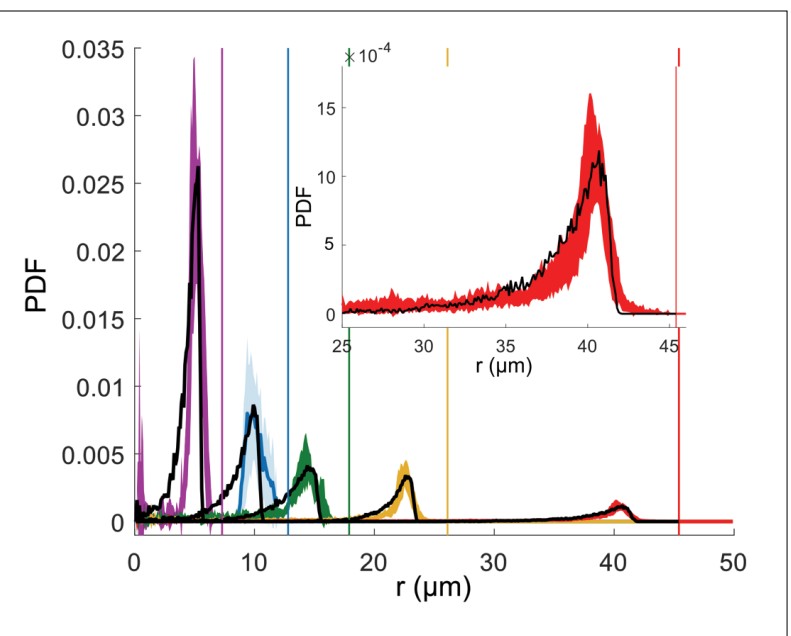

**Figure 2.** Mean radial distribution of the bacteria given as a probability density function (PDF) (filled line) with the corresponding standard deviation (colored area) for different microfluidic trap dimensions (trap radius 7.5 μm in violet, 12.5 μm in blue, 17.5 μm in green, 25 μm in yellow, and 45 μm in red), and the corresponding wall positions (vertical lines) in the absence of magnetic fields. Each curve is a mean over about six different traps. For small traps (7.5, 12.5, and 17.5 μm) with only 1 bacterium, and for 25 and 45 μm traps with the maximum number of 2 and 4 bacteria are examined, respectively. The total trajectory points acquired for each bacterium is about $10^4$. The black dotted lines show the corresponding simulated distributions. Inset shows an enlargement of the peak for the biggest microfluidic trap with radius of 45 μm.

The online version of this article includes the following figure supplement(s) for figure 2:

**Figure supplement 1.** Simulated three-dimensional (3D) trajectories without wall torque.

**Figure supplement 2.** Radial distribution without wall torque.

**Figure supplement 3.** $T_w$ and $A_w$ as function of the trap radius.

**Table 1.** Experimental peak positions at zero magnetic field.

To determine the peak position, the experimental mean distributions at zero magnetic field (without the long tail) was fitted with the MATLAB fitting tool with a Gaussian $a\exp\left(-\left(r-b\right)^2/c^2\right)$.

| Trap size | Trap radius (μm) | Peak position from center, $b$ (μm) | Width, $c/\sqrt{(2)}$ (μm) | Distance from wall (μm) |
|---|---|---|---|---|
| 1 | 7.3 | 5.1 | 0.5 | 2.2 |
| 2 | 12.8 | 10.0 | 0.8 | 2.8 |
| 3 | 17.9 | 14.3 | 0.9 | 3.6 |
| 4 | 26.1 | 22.5 | 0.6 | 3.6 |
| 5 | 45.4 | 40.3 | 0.8 | 5.1 |

both near the walls and in the trap interior. To check whether reversals of motion are influenced by the walls, we estimated the reversal rate from the number of observed events (*Figure 3*). Reversals occur with a rate of the order of 0.5 s⁻¹, that is, typically bacteria swim over distances exceeding the trap size before reversing their motion. The reversal rate shows some variability between experiments, but it does not show a systematic dependence on the trap sizes (*Figure 3—figure supplement 1*). For a direct test of the influence of the walls, we determined the reversal rate as a function of the distance of the wall. The resulting rate is noisy, in particular for the small traps, but does not show a systematic dependence on the distance from the wall (*Figure 3*). Specifically, we do not see an enhanced rate of reversals for bacteria in contact with (or in direct proximity to) the wall. These observations suggest that interactions with the walls do not lead to active changes of the swimming direction, but only result in alignment of the swimming direction along the wall.

Next, we turned to simulations to investigate the mechanism for motion along the walls. We used an ABP model that describes swimming bacteria as self-propelled particles subjected to rotational diffusion – a theoretical approach that has been used extensively to study microswimmers and other active particles in general (*Bechinger et al., 2016*; *Berdakin et al., 2013a*; *Berdakin et al., 2013b*; *Guidobaldi et al., 2015*; *Kalinin et al., 2009*; *Khatami et al., 2016*; *Perez Ipiña et al., 2019*; *Romanczuk et al., 2012*; *Takatori et al., 2014*), but also specifically for magnetic swimmers and magnetotactic bacteria (*Codutti, 2018*; *Telezki and Klumpp, 2020*). The simulation is 3D, with radii and depth to the trap taken from the experimental values. The motion is confined in quasi-2D as a result of the surface interactions (both ceiling, bottom and side walls). Using the measured value for the swimming speed, we systematically varied the interactions with the walls and compared the radial distributions from the simulations to the experimental ones. The wall interaction was tuned with two parameters: the interaction range $A_w$ (which gives an indication of the distance from the wall at which hydrodynamics acts and thus mainly influences the peak position) and the strength of an effective wall torque $T_w$ , which reorients the cells upon contact with the walls. In the absence of such wall torque ($T_w = 0$), the peak of the radial distribution is closer to the wall than observed in the experiments (*Figure 2—figure supplements 1 and 2*). Simulations resulting from the interactions parameters that give the best match to the experiments (see Materials and methods) are included as the black dotted lines in *Figure 2*. The fit of the simulated to the experimental distributions results in an overall good match. Minor discrepancies (experimental distributions broader toward the wall, simulated distributions broader toward the interiors) are attributed to imperfections in the wall, possible variation of the trap radius with height, and the inaccuracies of tracking near the wall. The fitted parameters $A_w$ and $T_w$ depend on the microfluidic trap size (thus on the curvature of the trap, see *Figure 2—figure supplement 3*), and are used for all following simulations at different magnetic fields but the same size. A similar wall torque was previously used to describe the swimming of algae (*Ostapenko et al., 2018*) and likely represents the reorientation of the cells due to steric as well as hydrodynamic interactions with the walls. While for algae the torque was derived from an asymmetric dumbbell model (*Ostapenko et al., 2018*), here we used a more general, phenomenological approach with an effective torque and two free fitting parameters $A_w$ and $T_w$ (see Materials and methods – 'Computational modeling' section) that is applicable independent of cell shape, swimming behavior, and the specifics of the corresponding hydrodynamics. While this phenomenological approach does not distinguish between hydrodynamic and steric origins of reorientation at the wall, a wall torque of the type we

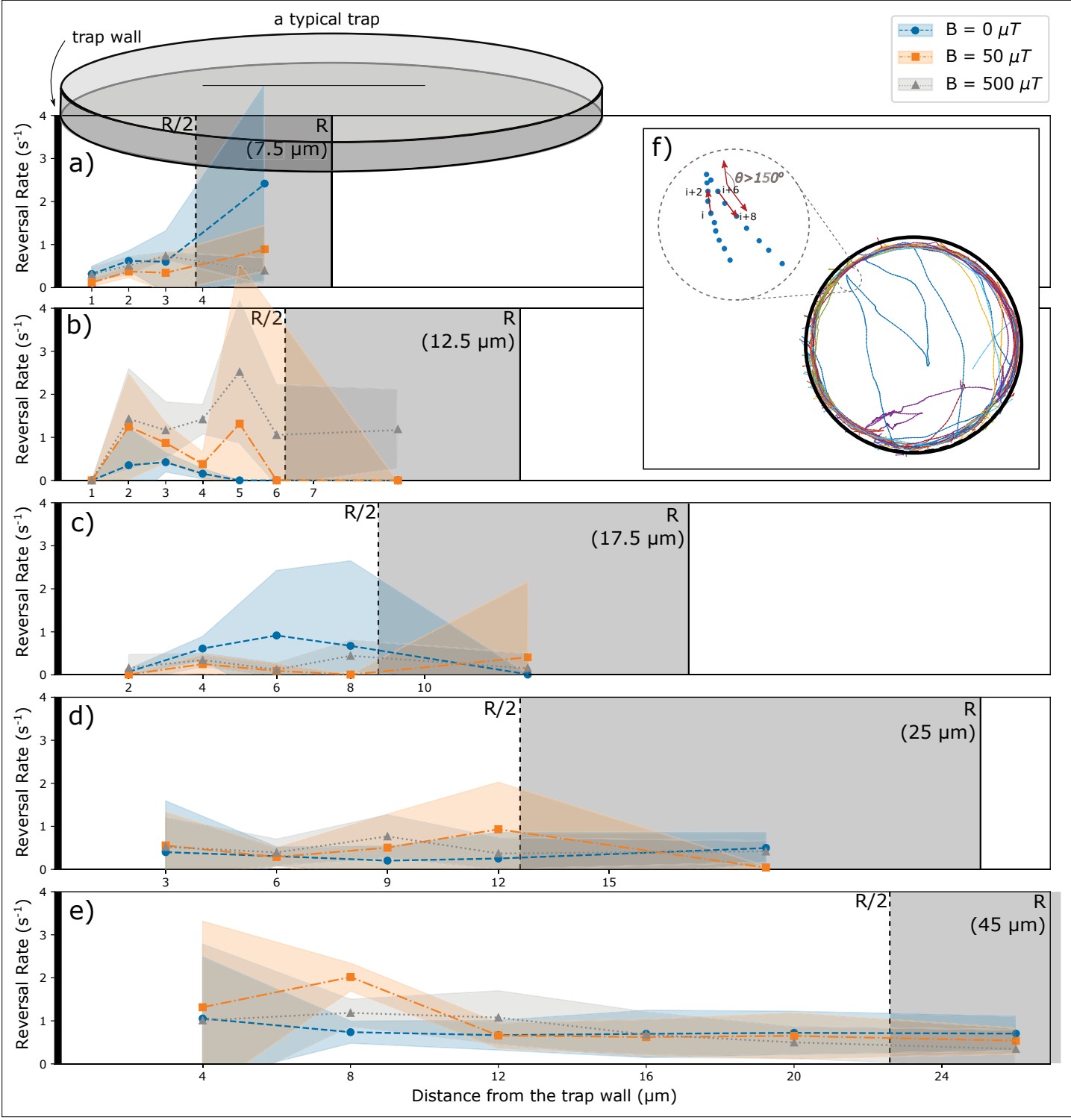

**Figure 3.** Analysis of reversals as a function of the distance from the wall. (**a–e**) Rate of reversals observed in traps of different sizes R, analyzed separately for different distance intervals from the wall. Sizes and number of intervals depend on the trap size. All distances from the walls between R/2 and R were pooled (trap interior) and are shown as the gray area (not to scale for the largest trap). Different colors show data for different strengths of the magnetic field. (**f**) Example trajectories showing reversals near the wall and in the interior. Reversals are identified by an angle exceeding 150° between the directions of motion at tracking point I and tracking point (i+6). Reversal rates are calculated as the number of reversal events per tracked time in the corresponding distance interval.

The online version of this article includes the following figure supplement(s) for figure 3:

**Figure supplement 1.** Reversal rates as a function of trap radius and magnetic field strength.

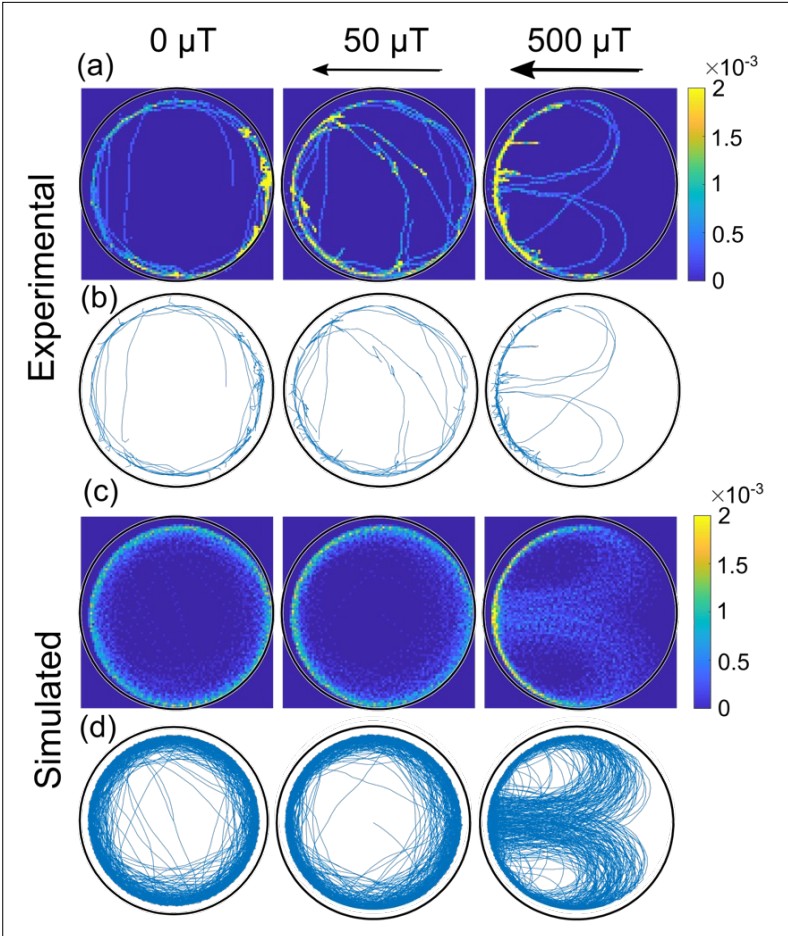

**Figure 4.** Comparison of experimental and simulated trajectories in a confined microenvironment. Heat-maps (rows a and c, 1 µm × 1 µm resolution; the color bar shows the normalized counts) and corresponding trajectories (rows b and d) for bacteria swimming with the magnetic north pole in front, at different magnetic fields (0, 50, and 500 µT) in the largest microfluidic trap with a radius of 45 µm. The rows a and b are experimental trajectories (see also *Video 2*) whereas c and d are simulated. Bacteria in both the simulation and experiment move at an average velocity of 40 µm s⁻¹.

The online version of this article includes the following figure supplement(s) for figure 4:

**Figure supplement 1.** Interplay between wall-interaction torque and magnetic torque.

used here can be derived from the steric interaction of an elongated or rod-shaped swimmer and a wall, see Materials and methods – 'Computational modeling' section. The shape of the radial distributions indicates that different swimmers behave similarly in strong confinement, and that this behavior can be effectively described with a general method that ignores the specificity of the hydrodynamics of the swimmer.

## Influence of a magnetic field

While the behavior described so far is similar to the previously observed swimming of algae in a circular environment (i.e. with the algae swimming at the wall), here a magnetic field can be used as an additional external control parameter in the case of magnetotactic bacteria. A magnetic field exerts a torque on these bacteria aligning their motion with the direction of the field. The observation of a torque resulting from the interaction with the wall thus suggests the possibility of competing effects, which we studied in both experiments and simulations. *Figure 4* shows how the magnetic field affects the trajectories in the largest microfluidic trap (here, for a bacterium swimming with its magnetic north pole at its front, the typical case in our experiments, as explained in Materials and methods – 'Selection of magnetic motile bacteria'). The magnetic field is oriented such that its north points to

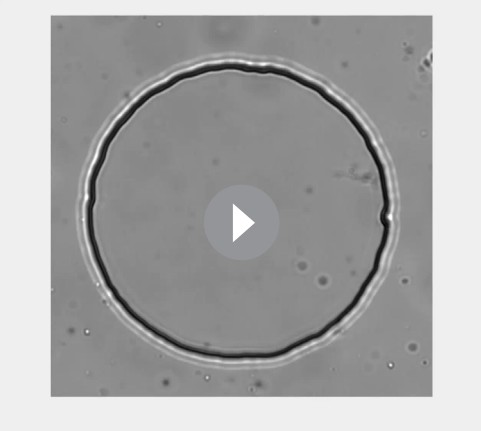

**Video 2.** Real-time magnetotactic bacteria (MSR-1) swimming in a microtrap with radius of 45 μm in the presence of external magnetic field (500 μT, horizontally, north pole directing to left). Imaged with 40× objective lens (NA = 0.6, air, Nikon) and an sCMOS camera (2560×2,160 pixels; Zyla, Andor Technology) with the rate of 50 fps.
https://elifesciences.org/articles/71527/figures#video2

the left side of the trap. For a field strength of 50 μT, comparable to the magnetic field of the Earth, the observed trajectories (both experimental and simulated) strongly resemble the one of the cases without magnetic field, and bacteria mostly swim along the wall circling the microfluidic trap. However, when a field of 500 μT is applied, we observe bacteria that perform U-turns inside the trap: that is a bacterium in the interior of the trap swims in the direction of the field until it hits the wall, where it continues its path along the wall, therefore turning against the direction of the magnetic field. Eventually, a U-turn realigns the bacterium with the field and moves it away from the wall and back into the interior of the trap. We note that U-turns are distinctly different from the abrupt reversals discussed above. These reversals do not show a dependence on the magnetic field, see *Figure 3*. In *Figure 4*, we also show the corresponding simulated trajectories (with the same velocity as the experiments, same microfluidic trap size, and with the parameters $A_w$ and $T_w$ determined from the mean radial distribution without magnetic field), which show the same behavior.

We understand the U-turns as reflecting the interplay between the wall torque and the magnetic torque (*Figure 4—figure supplement 1*). While the direction of the wall torque depends on whether the bacterium swims clockwise or counterclockwise (see Materials and methods – 'Computational modeling'), the magnetic torque depends on the magnetic moment orientation and on the swimming direction relative to the magnetic field. As an example, for a bacterium swimming clockwise with the north of its magnet at its front, the wall-interaction torque always points into the quasi-2D surface to which the bacteria are constricted to swim. At the magnetic north of the external field, the magnetic torque is opposite to the wall torque; at the south, the two torques add up (together with random noise) to direct the bacterium away from the wall and into the trap interior.

## Motility and cell-to-cell heterogeneity

Using our experimental setup, we observe a variety of different trajectories, even if we analyze bacteria that always swim with their north pole at their front and exclude the rare bacteria that exhibit directional reversals or swim persistently with their south pole at their front. To understand the variety, we simulated different trajectories with different strengths of the magnetic torque (dependent on $mB$, the product of the magnetic moment $m$ and the magnetic field $B$) and different swimming velocities $v$. The diagram in *Figure 5a* maps three simulated regimes, characterized by different types of trajectories for the largest microfluidic trap radius (45 μm): For very small swimming velocities, the bacteria are trapped at the north pole of the trap. For intermediate velocities, the bacteria perform the U-turns described above. Finally, for large swimming velocities, they circle around the trap. The transitions between these three behaviors depend on the field strength and can be understood by considering the interplay of different length scales. The transition between U-turns and circling depends on two length scales: the radius of the microfluidic trap $R_{\mathrm{trap}}$ and the radius of the U-turn $R_U$. The U-turn radius can be calculated from the balance between the magnetic torque and the rotational friction (*Esquivel and Lins De Barros, 1986*), which leads to

$$R_U = \frac{\pi}{2} \frac{\gamma_r}{mB} v = \frac{\pi}{2} \frac{k_B T}{mB D_r} v, \tag{1}$$

where $\gamma_r$ is the rotational friction coefficient, $D_r$ is the rotational diffusion coefficient, $k_B$ is the Boltzmann constant, and $T$ is the temperature. Thus, U-turns can only be observed if the U-turn radius is smaller than the microfluidic trap radius, otherwise the trajectories tend to follow the wall. These two

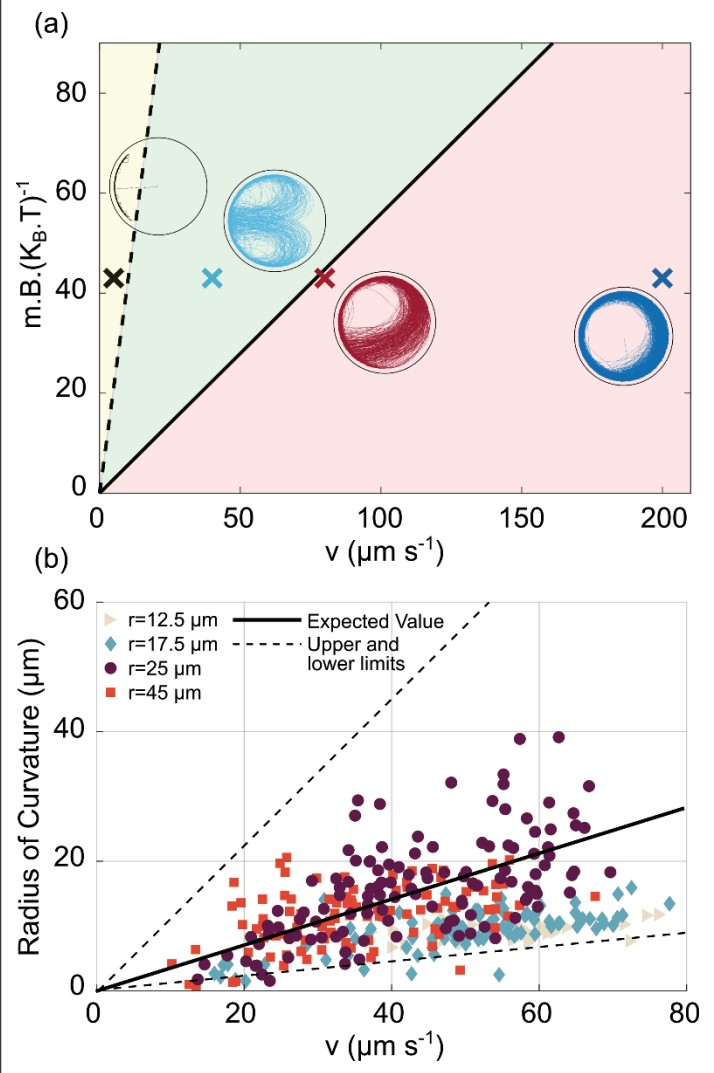

**Figure 5.** U-turn dependence on velocity and magnetic field. (**a**) Phase diagram of the trajectories in the space of $\left(v, \frac{mB}{k_BT}\right)$ for the largest microfluidic trap (radius 45 µm) with a magnetic field pointing to the left. The solid black line is given by *Equation 2* and represents the transition between the red area (for which U-turns are not visible since they are larger than the microfluidic trap radius), and the green area, for which the U-turns are clearly visible. The black dashed line is given by $\boldsymbol{R_U} = \boldsymbol{A_w}$ and represents the transition between the green area and the yellow area, where the trajectories tend to be confined at the wall since the U-turn radius is smaller than the wall-interaction-torque range. The points shown here are taken at $\frac{mB}{k_BT}$ = 43 given by the experiment ($B$=500 µT, $m$=0.36 × 10⁻³ A µm², $T$=305 K). The insets are the corresponding simulations of the trajectory for that region. (**b**) Experimental U-turn radius as a function of the measured velocity for experiments conducted at 500 µT. Different colors correspond to different microfluidic trap radii *r* (▶ 12.5 µm, ◆ 17.5 µm, ■ 25 µm, ● 45 µm, with no U-turns for the smallest microfluidic trap of radius 7.5 µm). The black line corresponds to the theoretical prediction $\boldsymbol{R_U} = \frac{\pi}{2} \frac{k_BT}{mBD_r} v$ with $m$=0.36 × 10⁻³ A µm² and $\boldsymbol{D_r}$ = 0.1 s⁻¹ for a typical micron size microswimmer; the dashed lines correspond to the magnetic moment of $\boldsymbol{m - \delta m}$ and $\boldsymbol{m + \delta m}$ which are the upper and lower limits for magnetic moment extracted from distribution of the size and number of magnetosomes in transmission electron microscopy (TEM) images (see Materials and methods – 'Magnetic moment measurement' for details).

The online version of this article includes the following figure supplement(s) for figure 5:

**Figure supplement 1.** Radii of U-turns vs. velocities for different microtrap sizes.

**Figure supplement 2.** Velocity distributions.

**Figure supplement 3.** Heterogeneity in magnetic moment.

regimes are indicated by the green and red areas in *Figure 5a*, respectively. The transition between these two behaviors is given by the black line, which represents the points where the U-turn radius is the same as the microfluidic trap radius,

$$\frac{mB}{k_BT} = \frac{\pi}{2}\frac{v/D_r}{R_{\text{trap}}}.$$ (2)

We note that this expression relates two dimensionless quantities, the ratio of the magnetic and thermal energies and the ratio of the persistence length of active motion (in the absence of a magnetic field, $v/D_r$) and the trap size. The transition to the north polar trapping regime (yellow area in *Figure 5a*) can also be understood by a comparison of length scales: if the U-turn radius is smaller than the interaction range $A_w$ of the wall, then the bacterium tends to remain on the wall, and spends most of its time at the north pole of the microfluidic trap. This transition is marked by the dashed line in *Figure 5a*.

The interplay between these length scales suggested by our theoretical analysis is confirmed by our experimental data. In *Figure 5b*, we plot the measured U-turn radius against the measured velocity, pooling data for all microfluidic trap dimensions, for an example case at 500 μT. The data points cluster around the theoretical prediction for the U-turns from *Equation 1* (solid line in *Figure 5b*), obtained using the mean magnetic moment estimated from transmission electron microscopy (TEM) images to be $m$=0.36 × 10⁻³ A μm² (see Materials and methods – 'Magnetic moment measurement'). The scatter around the theoretical expectation can be attributed to cell-to-cell variability in the magnetic moment, as discussed below. For the biggest microfluidic traps of radii 25 and 45 μm, U-turns are observed even for high velocities and low magnetic moments, whereas for smaller microfluidic traps 12.5 and 17.5 μm, only very small U-turn radii are observed and obviously none are bigger than the microfluidic trap size itself. Especially for higher velocities and small microfluidic traps, we can observe U-turns only for bacteria with higher magnetic moments (*Figure 5—figure supplement 1*).

The analysis in *Figure 5b* also indicates (and makes use of) the large cell-to-cell variability with respect to the swimming velocity and the magnetic moment. The large heterogeneity of swimming velocities is also shown by the histogram in *Figure 5—figure supplement 2*. The variability of the magnetic moment can be estimated from the distribution of the numbers and sizes of magnetosomes in TEM images (see Materials and methods – 'Magnetic moment measurement' and *Figure 5—figure supplement 3*), which results in a range $m \pm \delta m$ of magnetic moments and a corresponding range of U-turn radii, indicated by the two dashed black lines in *Figure 4b*. Notably, almost all measured U-turn radii fall into this range which further validates our model.

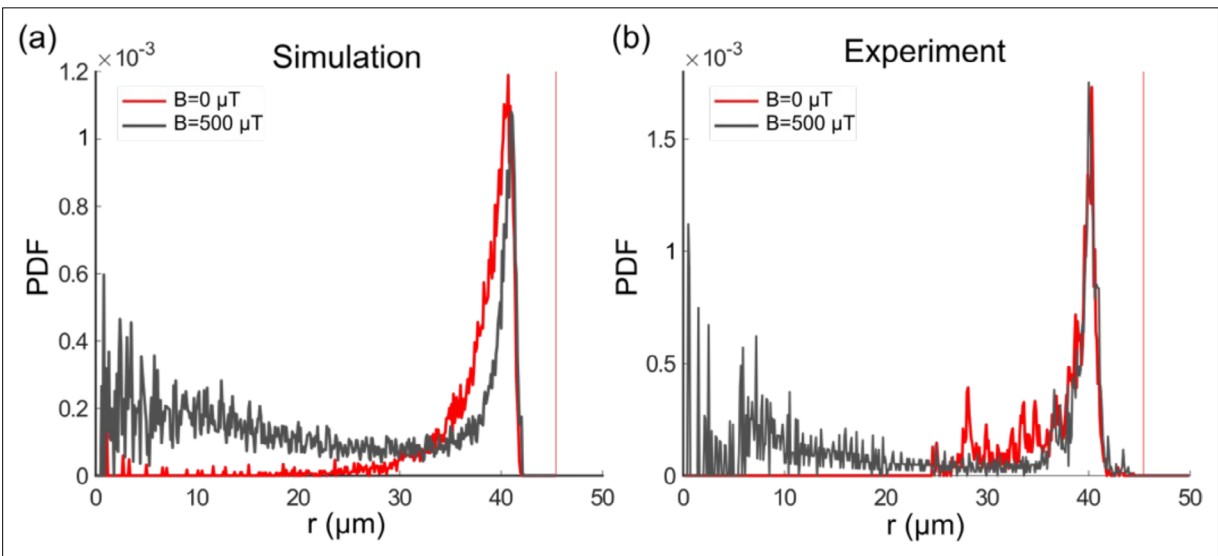

**Figure 6.** Single-cell radial distribution (probability density function [PDF]) in the absence of magnetic field (red) and with a magnetic field of 500 μT (gray) for the microfluidic trap size 45 μm, showing both the simulation (**a**) and experimental data (**b**), with a mean velocity of 40 μm s⁻¹. The vertical red dotted line represents the wall position.

The online version of this article includes the following figure supplement(s) for figure 6:

**Figure supplement 1.** The importance of single-cell analysis.

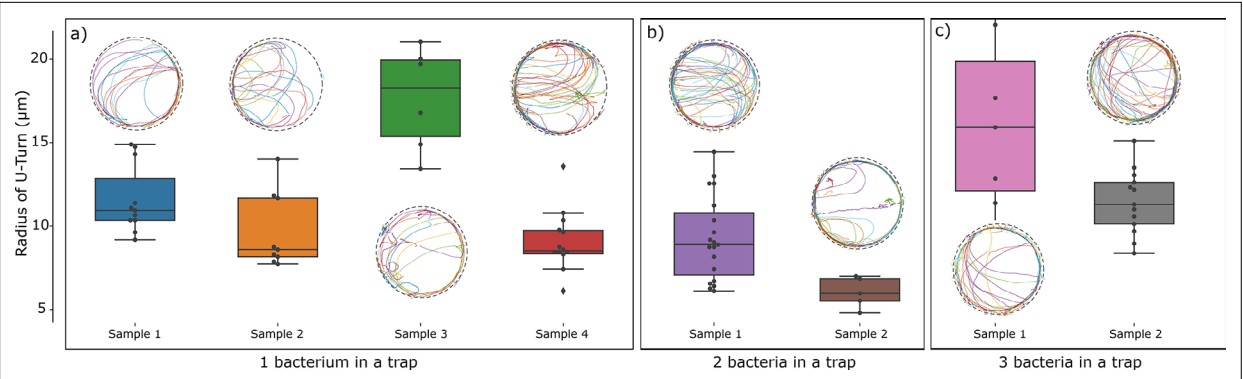

**Figure 7.** Test for interactions between bacteria: Comparison of trajectories and quantification of U-turn radii observed in traps containing one, two, or three bacteria. The comparison was possible for traps of radius 25 μm. Variability between trajectories reflecting bacteria heterogeneity is seen to dominate over interaction effects.

The online version of this article includes the following figure supplement(s) for figure 7:

**Figure supplement 1.** Trajectories of bacteria in traps (radius 25 μm) containing one, two, or three bacteria.

The U-turn behavior is also reflected in the radial distribution of the bacteria, where U-turns lead to a second, smaller peak toward the center of the trap that is absent without a magnetic field, as seen in both simulations and experiments (*Figure 6a and b*, respectively). Here, it is important to stress that this peak is not seen in the population-averaged distribution, but only for individual bacteria (see *Figure 6—figure supplement 1*), again due to the heterogeneity in bacterial parameters. Thus, single-cell analysis is crucial for a quantitative understanding of the mechanisms underlying these trajectories. We also note that the position and width of the main peak toward the exterior of the trap are unaffected by the magnetic field, indicating that the interactions with the wall (meaning, the parameters $A_w$ and $T_w$) are not influenced by the presence of a magnetic field.

We also made use of the U-turn radius to check for possible interaction effects between bacteria by comparing trajectories in traps (radius 25 μm) containing one, two, or three bacteria (*Figure 7*). The observed trajectories were qualitatively the same, independent of the number of bacteria in a trap for all three values of the magnetic field strength. To quantify possible differences, we analyzed the U-turn radii. The difference between the trajectories in traps containing different numbers of bacteria were smaller than the differences between different traps containing the same number of bacteria. We therefore conclude that effects of interactions between bacteria are small compared to the variability in U-turn radius due to bacterial heterogeneity (in their velocity and magnetic moment).

## Concluding remarks

In this study, we have investigated the swimming behavior of individual magnetotactic bacteria trapped in micron-sized compartments, in particular the interplay between their interaction with the confining walls and an external magnetic field guiding their motion. This unique combination of micro-sized confinement within a controllable magnetic field has allowed us to observe a variety of behaviors and to compare the data to a theoretical model. Two notable results emerged from these observations and from their theoretical analysis. On the one hand, we showed that the variety of observed behaviors could be explained by a rather simple physical description, based on the balance of two torques, the magnetic torque and a wall torque arising from the interaction with the trap walls. On the other hand, our observations reveal a large heterogeneity in the bacterial population with respect to parameters of the motility as well as the magnetic properties, in agreement with previous results (*Mitchell and Kogure, 2006*; *Stanton et al., 2017*; *Wang and Ford, 2009*).

As a result of that heterogeneity, the swimming behavior of these bacteria is blurred by observations at the population level, and single-cell observations and quantification are crucial for a full understanding of bacterial motility. The microfluidic trapping approach used in this work provides such quantitative single-cell characterization and may be extended to the quantification of other parameters of motility (e.g. rates of pausing or reversals, and chemotactic behaviors) as well as other processes including cell division and possibly flagellum assembly. Experimental challenges of this

approach include the trap realization and closure as well as the video analysis, which required stabilization. Our platform is able to provide a reproducible trap volume in which we can define the dimensions for comparison to the simulated data. This is in contrast to confinement within microfluidic droplets for example where the bacteria can swim in 3D (*Vincenti et al., 2019*) and imaging of the single trajectories can prove to be difficult.

In the absence of a magnetic field, the observed behavior of the trapped bacteria was seen to be rather similar to that observed for single-cell algae in an earlier study (*Ostapenko et al., 2018*). In particular, in both cases, circling around the wall perimeter was observed, indicating a similar interaction with the wall despite the hydrodynamic differences between the organisms, which depend on their different propulsion mechanisms. In particular, the algae used in that study, *C. reinhardtii*, are well established to be puller-type swimmers (*Drescher et al., 2010*). For the magnetotactic bacteria we studied here, they possess one flagellum at each pole, the swimmer type is not clear, but if their propulsion is dominated by the rear flagellum, they are expected to be pusher-type swimmers. The similarity in swimming behavior in confinement regardless of the propulsion mechanism might therefore indicate that steric effects dominate the interactions with the walls possibly through wall alignment. Indeed, a transient alignment parallel to the wall is expected for all elongated particles due to steric interactions (*Elgeti and Gompper, 2009*; *Li and Tang, 2009*). However, the details of that interaction depend on the shape of the swimmer including its motility apparatus and, in some cases, seem to be dominated by the interaction of flagella with the wall (*Kantsler et al., 2013*). Nevertheless, we suspect that two ingredients that are relatively insensitive to the shape are sufficient to determine the transient wall alignment: an initial torque aligning the swimmer with the wall, which may result from the steric interaction of the front of the swimmer with the wall, and a mechanism for leaving the wall, which may be due to rotational diffusion, as swimmers leave the wall, once their orientation points away from it. We expect that the general picture of collisions with a wall will be rather similar when these two conditions are met, even though the detailed mechanisms underlying these collisions may differ between different types of microswimmers. Indeed, more detailed studies of the interactions with the walls or comparisons with other types of microswimmers will be needed to draw definite conclusions.

In contrast to algae, however, the magnetotactic bacteria can be steered 'remotely' with a magnetic field: one of the reasons why they are attractive candidate microswimmers for biomedical and environmental applications. Our observations show that steering magnetotactic bacteria needs to take the interactions with channel walls or obstacles into account, a situation expected to be rather frequent in application scenarios. However, they also show that if interactions with walls are taken into account, the interplay of simple physical forces can quantitatively explain and predict the swimming behavior. It will be interesting to generalize this result to other types of biological and synthetic magnetic microswimmers. For example, a completely different system of synthetic rollers confined in a circular geometry and subjected to flows (*Morin and Bartolo, 2018*) shows similar behavior to the one portrayed in this study, hinting at the flexibility of this approach.

In summary, we have shown how magnetic torques and torques due to interactions with confining walls direct the motion of bacteria in microfluidic traps, which provide a way to quantitatively address the individuality of the bacteria and thereby to obtain a more complete understanding of the physics and biology of their swimming. The study of the swimming in a defined structure will be useful for the future understanding of their behavior in more complex systems, including the porous environment in the sediment habitats or in biomedical application scenarios. Likewise, the microfluidic trapping approach may also be a useful tool for studying other species of microorganisms and synthetic microswimmers as well as other biological processes.

## Materials and methods
### Fabrication of the master molds and microfluidic systems
Two silicon wafers were used as master molds for the final PDMS chips: one for the upper control layer and one for the lower fluidic layer. The silicon wafers were initially baked at 200°C for 20 min and were allowed to cool down to room temperature. For the control layer, an SU-8 3010 (Micro-Chem Inc) thin film was spin-coated onto the wafer to a height of 20 µm, baked and exposed to UV light through a film mask (Micro Litho) with a mask aligner (Kloé UV-KUB 3) according to the

**Table 2.** Master mold fabrication parameters.
SU8 3010 spin-coating, baking, and UV exposure parameters used for the fabrication of the master molds.

| CONTROL LAYER (final height: 20 µm) | |
|---|---|
| Spin-coating | 15 s 500 rpm, 30 s 925 rpm |
| Softbake | 3 min 65°C, 10 min 95°C, 1 min 65°C |
| Exposure | 5 s |
| Post-exposure bake | 1 min 65°C, 6 min 95°C, 1 min 65°C |
| FLUIDIC LAYER (final height: 10 µm+10 µm) | |
| Spin-coating | 15 s 500 rpm, 30 s 3000 rpm |
| Softbake | 1 min 65°C, 3 min 95°C, 1 min 65°C |
| Exposure | 5 s |
| Post-exposure bake | 1 min 65°C, 3 min 95°C, 1 min 65°C |
| Spin-coating | 15 s 500 rpm, 30 s 4,000 rpm |
| Softbake | 1 min 65°C, 3 min 95°C, 1 min 65°C |
| Exposure | 5 s |
| Post-exposure bake | 1 min 65°C, 3 min 95°C, 1 min 65°C |

manufacturer's recommendations. For the fluidic layer, multilayer photolithography was employed. First, an initial SU-8 3010 10 µm film was spin-coated onto the wafer, baked and exposed according to the manufacturer's recommendations. Then, another SU-8 3010 10 µm film was spin-coated, baked and exposed (with a second mask) again according to the manufacturer's recommendations, but with a slight variation in the spin-coating parameters, where the spin speed was increased to achieve a 75% of the target height resulting in a height of 10 µm. The wafers were then developed with mr-Dev 600 (microresist technologies). A more detailed description of the wafer fabrication parameters can be found in *Table 2*. The SU-8 film height was measured with a white light interferometer (Wyko NT1100). Prior to their use, the master molds were treated with 1*H*,1*H*,2*H*,2*H*-perfluorodecyltriethoxysilane 97% (abcr) to reduce PDMS adhesion upon usage.

The final two-layered microfluidic device was produced with (PDMS) via soft lithography. Briefly, PDMS elastomer monomer and curing agent (Sylgard 184, Dow Corning) were mixed in a ratio 10:1 and then degassed. For the fabrication of the control layer, the PDMS was cast onto the master mold to a height of about 5 mm and cured at 80°C for 2 hr. For the fluidic layer, the PDMS was spin-coated onto the master mold until reaching a thickness of 40 µm (spinning at 500 rpm for 30 s, then 2000 rpm for 60 s), and cured at 80°C for 1 hr. The inlets for the control layer were then punched with a 1 mm diameter biopsy punch (pmfmedical). The control and fluidic layers were plasma treated, aligned, and then bonded. Subsequently, the fluidic layer inlet and outlet were punched with a 1.5 mm diameter biopsy punch. Finally, the PDMS assembly and a clean glass coverslip were bonded by plasma activation to finish the microfluidic device (*Figure 1—figure supplement 2*).

## Operation of the chip

The control and fluidic layers were filled by centrifugation for 10 min at 900 RCF with MSR-1 growth medium or milliQ water. The control layer was connected to a pressurized nitrogen source connected through a series of silicon tubing (ID=1 mm, Roth) to a custom-built valve device (*Kubsch et al., 2017*). The silicon tubing was inserted into the microfluidic inlets by means of a metal connector. The fluidic layer was connected by a metal connector to a PFTE tubing (ID=0.8 mm, Saint-Gobain Performance Plastics Isofluor GmbH) which connected the device to a 1 mL plastic syringe (Norm-Ject, Henke-Sass Wolf) filled with a bacterial suspension or with calcein. The syringe was loaded onto a mechanical syringe pump (Aladdin, WPI) and liquid was dispensed into the microfluidic system at a flow rate of 5 µL min⁻¹. For closing the microfluidic traps, the valves were actuated by applying a pressure of 1.75 bar with nitrogen (*Figure 1—figure supplement 2*).

## Selection of magnetic motile bacteria

MSR-1 was cultured in MSR-1 growth medium with the composition indicated by *Heyen and Schüler, 2003*, with the addition of pyruvate (27 mM) as carbon source instead of lactate. For the creation of an aerotactic band and the subsequent selection of swimming bacteria, the growth medium was supplemented with 0.1% agar. Briefly, 1 mL of bacteria was inoculated into the bottom of a 15 mL Hungate tube filled with 10 mL of MSR-1 growth medium with 0.1% agar. The tube was sealed with a rubber cap pierced with a needle capped with a 0.2 µm filter (Whatman) to allow the formation of an oxygen gradient. To have the north-seeking bacteria (i.e. swimming with the north of their magnet at their front in oxic conditions) in the formed band, the tube was put inside a pair of coils to apply a magnetic field parallel to the oxygen gradient, but pointing downward to the anoxic region (equivalent to the situation that the bacteria experience in their natural habitat in the Northern Hemisphere). The bacteria were grown at 28°C and allowed to form an aerotactic band. Once the band had formed, the motile bacteria were selected by harvesting the band and culturing them for two passes in standard MSR-1 growth medium and microaerobic conditions. Subsequently, the magnetic bacteria were collected with a needle by placing a magnet next to the tube to attract them. The final population of north-seeking bacteria was estimated to be about 80% of the population. In oxic conditions as in our experiment (where oxygen can freely diffuse through the PDMS into the trap), these bacteria swim with the north of their magnet at their front. Nevertheless, few traps contained south-seekers, swimming with the south of their magnet at their front, and thus they were excluded from the subsequent analysis. The optical density was measured at 565 nm ($OD_{565}$) and was adjusted as needed with fresh growth medium for subsequent measurements.

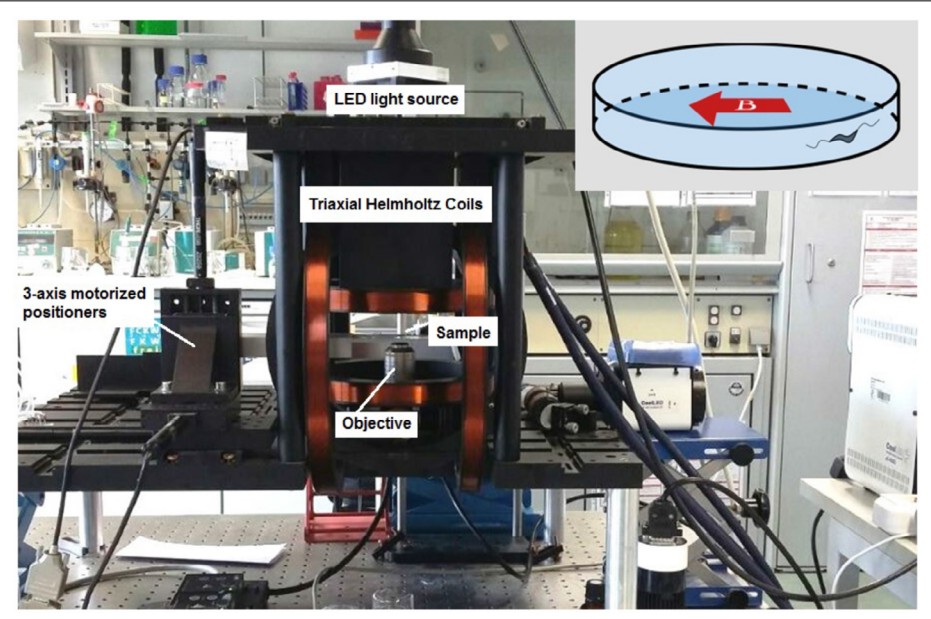

**Figure 8.** Magnetic microscope. An inverted optical microscope to take images using a long working distance 40× objective lens. A white light LED was used as a light source to illuminate the sample. The microscope was set up inside 3D-axis Helmholtz coils with a controller with a precision of ±2.5 µT. It was also equipped with three motorized linear stages to provide sample positioning in three dimensions (3D). Inset: The direction of the applied magnetic field in experiments; the magnetic field was set to 0, 50, or 500 µT parallel to the microtrap diameter.

The online version of this article includes the following figure supplement(s) for figure 8:

**Figure supplement 1.** Image stabilization process for effective background subtraction and bacteria tracking.

**Figure supplement 2.** Three-dimensional (3D) trap reconstruction.

**Figure supplement 3.** Sample U-turn of bacteria in the microtrap.

**Figure supplement 4.** Sample trajectory of bacteria and U-turn.

**Figure supplement 5.** Theoretical wall-interaction scheme.

## Microscopy and image stabilization

Bacteria confined within the microfluidic traps were imaged on a custom inverted microscope (**Bennet et al., 2014**) using a long working distance 40× objective lens (NA = 0.6, air, Nikon) and an sCMOS camera (2560×2160 pixels; Zyla, Andor Technology). The sample was illuminated with white light by an LED illumination system pE-4000 (CoolLED Ltd, Andover, UK). Images were captured with Andor iQ3 software at a pixel size of 111 nm pixel$^{-1}$, and saved as a 16-bit tiff stack. Each stack consisted of 2000 slices per experiment, with 20 ms time step between slices.

The microscope was equipped with 3D-axis Helmholtz coils with a controller (C-SpinCoil-XYZ, Micro Magnetics Inc). These coils were used to apply DC magnetic fields with a precision of ±2.5 µT (5% of the Earth's magnetic field). A photo of the microscope platform and the 3D-axis Helmholtz coils is shown in **Figure 8**. A three-axis magnetic sensor (Micro Magnetics Inc) was used to measure the precise value of the Earth's magnetic field in sample location. The magnetic field was controlled using a LabVIEW (National Instruments) based program and the ambient magnetic field was canceled out before each experiment. The magnetic field was then set to 0, 50, or 500 µT parallel to the microfluidic trap diameter (inset of **Figure 8**). The microscope was equipped with three motorized linear stages (Z825B, Thorlabs) to provide sample positioning in 3D.

Due to the micron size of the microfluidic traps and the fact that the bacteria tend to swim near the walls, even a small amount of vibration hinders correct background subtraction and consequently the bacteria tracking process. Therefore, a custom-made script was developed in MATLAB (Math-Works, Inc) for image stabilization and background subtraction (**Charsooghi et al., 2021**). The image stabilization was based on comparing two successive frames and detecting the corners and features that were sharp and changed drastically (the green crosses in **Figure 8—figure supplement 1**). Then, matched points were detected and connected, and transformation (including scaling, rotation, translation, and shearing) was applied to the image to overlay the points (**Figure 8—figure supplement 1e,f**). An effective background subtraction, and hence tracking of the bacteria, could be achieved after image stabilization (**Figure 8—figure supplement 1g,h**).

## Microfluidic trap characterization

A total number of five different microfluidic trap dimensions with radii of 7.5, 12.5, 17.5, 25, and 45 µm are used. To determine the actual microfluidic traps dimensions, the microfluidic traps were imaged with a confocal laser scanning microscope (SP8 DMi8, Leica) equipped with 63× objective (water immersion, NA=1.2, Leica). For this purpose, the fluidic and control layers were initially filled with milliQ water. The water in the fluidic layer was then exchanged with a fluorescent solution containing 25 µM of calcein and 10 mM HEPES at pH 7.0, and the valves were subsequently actuated to close the traps. For each size, 10 microfluidic traps were imaged and z-stack including 50 z positions were recorded with a 488 nm laser with a z-stack height of 356 nm per microfluidic trap. The microfluidic trap radii were determined by fitting a circle to the fluorescent area (**Figure 8—figure supplement 2**, and **Table 3**). The height was determined by multiplying the number of z-stacks bearing a fluorescent signal by the z-stack height.

## Bacterial imaging

Initially, the control layer and the fluidic layer of the microfluidic device were filled with MSR-1 growth medium. A bacterial suspension with the required $OD_{565}$ was then loaded into the microfluidic device

**Table 3.** Measured microtrap diameter.

Extracted mean microtrap diameters from fitting a circle to the fluorescent area of the microtrap *x-y* slice. The error is calculated by the standard deviation of the measured diameter over different z-stacks and different examined microtraps (*n*=11 for each microtrap size). The measured values are in good agreement with the nominal values of the printed mask patterns.

| | Size 1 (µm) | Size 2 (µm) | Size 3 (µm) | Size 4 (µm) | Size 5 (µm) |
|---|---|---|---|---|---|
| **Diameter** | **15** | **25** | **35** | **50** | **90** |
| $D_{measured}$ | 14.85 | 26.52 | 35.8 | 52.13 | 90.88 |
| $D_{error}$ | 1.04 | 0.44 | 0.42 | 0.65 | 1.05 |

at a flow rate of 5 μL min⁻¹. Once the bacterial suspension had reached the fluidic channels (under the traps), the flow was stopped and the valves were lowered with 1.75 bar pressure, thus confining bacteria in the closed microfluidic traps. The experiments were conducted with all available five different microfluidic trap dimensions (subsection: Microfluidic trap characterization). Each experiment was recorded for 40 s (2000 frames with 50 fps). The experiments were repeated five to eight times independently, with different microfluidic traps and bacteria for each trap size, which corresponds to roughly $10^4$ total trajectory points. The final number of bacteria inside the microfluidic trap can be tuned by controlling the initial concentration of the bacterial suspension (*Figure 1—figure supplement 3*). For the trap sizes of radius 7.5, 12.5, and 17.5 μm, microfluidic traps with only one bacterium were selected to avoid bacterium-bacterium interactions. However, for traps of radius 25 and 45 μm, the videos were recorded with a maximum of three and four bacteria per trap, respectively. The microfluidic traps were opened and washed out every 10 min by fresh medium and bacteria.

Prior to each measurement, the magnetotactic behavior of the bacteria was inspected by applying a rotating magnetic field of 1 mT to the sample. Only bacteria showing a magnetic field-dependent response were recorded. The rotating magnetic field was shown not to alter the posterior swimming behavior of the bacteria.

## Bacterial tracking

Bacteria tracking in 2D was performed using custom-made scripts in MATLAB, after image stabilization and background subtraction (see *Figure 8—figure supplement 3*). The bacteria were identified based on an intensity threshold and their locations were determined by finding a centroid of contiguous pixels in a region with intensity above this level. The trajectories were formed by connecting centroid positions in subsequent frames. Resulting trajectories were smoothed with a five-point moving average (*Figure 1—figure supplement 4* and *Video 1*). The points resulting from the incorrect tracking of the wall were discarded by applying a cutoff circle with radius of 1.1 times the radius of the trap according to the thickness of trap wall shadow in the recorded images (see *Figure 1—figure supplement 4*). For the trap with 7.5 μm radius, the tracks were obtained by hand through the help of a self-written MATLAB code, since the automatic tracking gave poor results. The instantaneous fourth-order velocities along the trajectories were calculated for all trajectories. The experiments at a given condition (radius and magnetic field) were repeated for six to eight different microfluidic traps, each containing different bacteria.

## Data analysis

The trajectories, both experimental and simulated, are processed and analyzed in the same way. In particular, the U-turn radii, the distribution of the bacteria, and the heat-maps were extracted as follows.

### U-turns

The U-turns are identified through a semi-manual algorithm adapted from *Thageman, 2020*, to measure the U-turn radius, a circle is fitted on the manually selected portion of the track presenting a U-turn. The mean U-turn radius is then obtained for each trap. For the experimental trajectories only, the corresponding velocity of the bacterium when performing the U-turn was measured by averaging over instantaneous velocities in that segment of the trajectory (*Figure 8—figure supplement 4*). Sample U-turns in different traps sizes in the presence of 500 μT of magnetic field are shown in *Figure 8—figure supplement 5*.

### Radial distributions

To calculate the radial distribution, the trap is divided into concentric rings of width $\Delta r = 0.1$. Then, the radial distribution function of the trap occupancy is determined by counting the number $n$ of frames where a bacterium is in a shell with thickness of $\Delta r$ at a distance of $r$ from the center of the trap, and averaging over the total number of bacteria in different traps of the same size $N$, resulting in a probability density function (PDF) of $n/(N2\pi r\Delta r)$ (*Ostapenko et al., 2018*). The distribution for each value of magnetic field and trap size was obtained as a mean over the single traps, and the error is given by one standard deviation.

## Heat-map

To visualize the high-density points in the traps, a heat-map was produced by binning areas of 1 µm × 1 µm, calculating the number of points of the tracks in the bin, and normalizing by the bin area and the total number of points.

## Computational modeling

The simulation is based on an ABP method in 3D (*Codutti et al., 2019*). Bacteria are represented by spheres, whose single trajectories are simulated by integrating the following equations:

$$\gamma_t \frac{dr}{dt} = \gamma_t v e + F_{WCA} + \sqrt{2 k_B T \gamma_t} \xi_t \tag{3}$$

$$\gamma_r \frac{de}{dt} = \left[ m \times B + T_{\text{reor.}} + \sqrt{2 k_B T \gamma_t} \xi_r \right] \times e, \tag{4}$$

where $t$ is the time, $r$ is the position of the bacterium, $e$ is the direction vector, $T$ is the temperature, $k_B$ is the Boltzmann constant, $\gamma_t$ and $\gamma_r$ the translational and rotational friction coefficients, respectively, $v$ is the speed of self-propulsion, $m \times B$ is the the external magnetic torque with $m$ being the magnetic moment of the bacterium and $B$ the external magnetic field, $F_{WCA}$ and $T_{\text{reor.}}$ are the force and reorientation torque due to the interaction with the wall (see the following paragraph), and $\xi_t$ and $\xi_r$ describe uncorrelated white noise in the translational and rotational degrees of freedom.

The trap is composed of a lateral curved wall and two flat surfaces (bottom and ceiling), with dimensions matching those of the experimental ones. The wall interaction is implemented as follows (see *Figure 8—figure supplement 5*). An imaginary sphere of radius $A_w$ is positioned with its center on the wall. The bacterium, represented as a sphere of radius $a$, interacts with the imaginary sphere and therefore is repelled by a Weeks-Chandler-Andersen (WCA) potential potential (*Ostapenko et al., 2018*; *Weeks et al., 1971*), which gives a force perpendicular to the wall and directed toward the center of the trap:

$$F_{WCA} = \begin{cases} -24\epsilon \frac{\hat{r}}{|r|} \left[ 2 \left( \frac{\sigma}{|r|} \right)^{12} - \left( \frac{\sigma}{|r|} \right)^6 \right] & \text{for } |r| < \sigma 2^{1/6} \\ 0 & \text{otherwise} \end{cases} \tag{5}$$

where $\varepsilon$ is the strength of the force set to $10^{-10}$ (to avoid unphysical interactions such as bacteria entering the walls), $r$, $\hat{r}$, $|r|$ are respectively the vector between the center of the imaginary sphere and the center of the bacterium $(x_{\text{im. sphere}} - x_{\text{bact.}})$, its unit vector and its modulus, and $\sigma = \frac{a + A_w}{2^{1/6}}$ with $a$ the radius of the bacterium. Moreover, upon interaction with the wall, the bacterium feels a reorientation torque:

$$T_{\text{reor.}} = T_w e \times F_{WCA} \, , \tag{6}$$

where $T_w$ is the strength of the torque. Hydrodynamics is not considered explicitly in the model, but is included indirectly in the effective reorientation torque. The free parameters are $A_w$ and $T_w$, which depend on the radius of the trap (thus on the curvature) but not on the magnetic properties of the system. These parameters are determined by comparison with the experiments as illustrated in the Materials and method section 'Fitting of the radial distribution'. We emphasize that the orientation vector $e$ is free to vary in 3D, but due to the small heights of the traps studied here, the interaction with the top and bottom walls results in alignment parallel to those walls most of the time, making the system quasi-2D also for the orientation vector.

We consider the description of reorientation of the bacteria at the wall by a wall torque as a phenomenological description, which does not distinguish between hydrodynamic and steric interactions with the walls and which can be used independently of the microscopic details of the interaction. Nevertheless, we want to point out that such a torque can be motivated from the steric interaction of a rod-shaped particle with a wall. Since our MSR-1 bacteria have an elongated (spiral-shaped) cell body and flagella at the front and rear poles, they can be approximated as rod-shaped object. If a rod-shaped bacterium is discretized as a chain of spherical particles, the force and torque it experiences from interaction with the wall can be expressed as sums over the particles, specifically the torque is given by $T = \sum (r_i - r) \times F_i$, where $r$ is the center of mass of the rod and $r_i$ and $F_i$ are the position of the individual particle in the rod and the steric wall force this particle experiences. The orientation or self-propulsion direction $e$ is the same for all particles in a rigid rod and $(r_i - r)$ is proportional to $e$,

**Table 4.** Simulation parameters.

The trap sizes are taken from the experimental measurements with fluorescent microscopy (see *Table 3*). The velocities are obtained from the mean experimental values with no magnetic fields (see *Figure 4—figure supplement 1*). The hydrodynamic parameters $T_w$ and $A_w$ are obtained by choosing the set of parameters for which the adjusted $R$ squared ($R^2_{adj}$) value between simulated and experimental data is minimal (see text). The errors (indicated as $\Delta^-$ and $\Delta^+$, respectively, for the lower and upper bound) are obtained varying one parameter at the time (keeping the other constant), calculating the $R^2_{adj}$, and allowing it to be 10% less than the maximum.

| Size | Trap radius (µm) | Velocity (µm s⁻¹) | $T_w$ (adim.) | $\Delta^-_{Tw}$ | $\Delta^+_{Tw}$ | $A_w$ (µm) | $\Delta^-_{Aw}$ (µm) | $\Delta^+_{Aw}$ (µm) | $R^2_{adj}$ |
|---|---|---|---|---|---|---|---|---|---|
| 1 | 7.3 | 30 | 14 | -1 | 1 | 2 | -1 | 1 | 0.70242 |
| 2 | 12.8 | 40 | 11 | -1 | 1 | 3 | -1 | 1 | 0.50831 |
| 3 | 17.9 | 45 | 9.5 | −1.4 | 1.1 | 3.5 | −0.7 | 0.3 | 0.83811 |
| 4 | 26.1 | 30 | 4.5 | -1 | 1.9 | 3.7 | −0.3 | 0.2 | 0.89411 |
| 5 | 45.4 | 40 | 2.8 | −0.7 | 1.1 | 6 | −0.4 | 0.3 | 0.89615 |

that is, $(\mathbf{r}_i - \mathbf{r}) = a_i\mathbf{e}$ (for a spirillum, this holds on average due to the rotation of the cell body). For a flat (or approximately flat) surface, the forces $\mathbf{F_i}$ are all parallel (approximately parallel) to the force on the center of mass, so $\mathbf{F}_i = b_i\mathbf{F}_{WCA}$, from which we obtain *Equation 6* with $T_w = \sum_i a_i b_i$. In a collision of a bacterium with a wall, this sum will be dominated by the terms corresponding to the tip of the approaching bacterium, as the force decays rapidly with distance from the wall.

## Fitting of the radial distribution

In the simulation, there are two free parameters: the interaction range $A_w$ (which gives an indication of the distance from the wall at which hydrodynamics acts and thus mainly influences the peak position) and the strength of a wall torque $T_w$. These free parameters are changed until the simulated curve matches the experimental radial distribution, and the final parameters (*Table 4*) are the ones minimizing the adjusted $R$ squared between the experimental and simulated curve (written in MATLAB, Mathworks Inc) $R^2_{adj} = 1 - \frac{n-1}{\nu}\frac{\sum(exp-sim)^2}{\sum(exp-\overline{exp})^2}$, where $n$ is the number of points and $\nu=2$ is the number of free parameters, *exp* is the arithmetic mean of the experimental data and *sim* is the simulated data. While the match is good for bigger microfluidic traps (with an adjusted $R$ squared of 0.896 for the microfluidic trap with radius of 45 µm, see inset of *Figure 2*), it gets poorer for smaller microfluidic traps (0.702 for the microfluidic trap with radius of 7.5 µm where manual tracking is used and 0.508 for the microfluidic trap with radius of 11.5 µm where the automatic tracking is used), due to the right-hand tail of the distribution of the experimental data. For the small microfluidic traps, the distribution is broader on the wall side of the peak, possibly due to irregularities of the overall trap shape of the microfluidic trap and because of slight variations in the microfluidic trap radius, the effects become more prominent for small traps. Excluding the smallest trap where manual tracking was used, the right-hand tail could be increased by artifacts of the algorithm which recognizes bright parts of the wall as a bacterium.

## Bacterial magnetosome counting

The number and diameter of the magnetosomes were computed from the TEM 2D images using a semi-automatic custom-made application in MATLAB. For each image, the user had to manually draw a region of interest (ROI) to delimit the boundary of each bacteria. These ROIs were used to count the magnetosomes separately for each bacterium and to avoid artifacts that could be found outside the cell. A build-in MATLAB function based on circular Hough transform (*Atherton and Kerbyson, 1999*) was used to automatically find the number of magnetosomes and their sizes. Manual filtering of artifacts was possible when the automatic segmentation failed.

## Magnetic moment measurement

The magnetic moments of the bacteria were determined by examining the number and diameter of magnetosomes inside the bacteria by TEM. TEM grids were prepared by adding 20 µL of bacterial suspension and letting them settle down during 30 min before washing them twice with milliQ water and drying them with paper. TEM images were taken with a transmission electron microscope (EM912, Zeiss) at an accelerating voltage of 120 kV with a magnification of 4000–10,000×.

The variability of the magnetic moment can be estimated from the distribution of the numbers and sizes of magnetosomes in the TEM images. According to the data of TEM images, the mean value of the number of magnetosomes inside bacteria is $N$=23.8 ± 8.3 and the mean size of magnetic nano crystals is $R$=19.4 ± 4.3 nm (*Figure 5—figure supplement 3*). The magnetosome diameters were calculated by fitting a circle to the magnetosome TEM images. Therefore, the mean value and the error of the magnetic moment are:

$$M = N\frac{4}{3}\pi R^3 \rho_M = 3.54 \times 10^{-16} Am^2 \tag{7}$$

$$\delta M = |M| \sqrt{\left(\frac{\delta N}{N}\right)^2 + \left(3\frac{\delta R}{R}\right)^2} = 0.75M \tag{8}$$

Here, $\rho_M$ is magnetic moment per unit volume of magnetite which is equal to $4.8 \times 10^{-22}\frac{A.m^2}{nm^3}$ (*CRC Handbook of Chemistry and Physics, 2005*). $N$ is the average number of magnetosomes inside magnetosome chain with the standard deviation of $\delta N$. $R$ and $\delta R$ stand for the mean radius of magnetosomes and the corresponding error, respectively. As the number of magnetosomes in the chain and the magnetosomes sizes are widely distributed, we calculate the highest and lowest possible magnetic moment of the chain. Based on *Equations (7) and (8)*, the expected value for the magnetic moment of the bacteria is included between the values $\pm 0.75M$.

## Acknowledgements

The authors thank Heike Runge for taking the TEM images. This work was supported by the Deutsche Forschungsgemeinschaft within the Priority Program 1726 on microswimmers (grant No. FA 835/7-2 and KL 818/2-2 to DF and SK). TR acknowledges support from the MaxSynBio consortium which is jointly funded by the Federal Ministry of Education and Research of Germany and the Max Planck Society. AC, ECD, and HMT were supported by the IMPRS on Multiscale Biosystems. The funders had no role in study design, data collection and analysis, decision to publish, or preparation of the manuscript.

## Additional information

### Funding

| Funder | Grant reference number | Author |
| --- | --- | --- |
| Deutsche Forschungsgemeinschaft | KL 818/2-2 | Stefan Klumpp |
| Deutsche Forschungsgemeinschaft | FA 835/7-2 | Damien Faivre |
| BMBF and Max Planck Society | MaxSynBio | Tom Robinson |
| IMPRS on Multiscale Biosystems | | Agnese Codutti Elisa Cerdá-Doñate Hubert M Taïeb |

The funders had no role in study design, data collection and interpretation, or the decision to submit the work for publication.

### Author contributions

Agnese Codutti, Conceptualization, Software, Formal analysis, Investigation, Writing - original draft, Writing - review and editing; Mohammad A Charsooghi, Formal analysis, Investigation, Writing

- original draft, Writing - review and editing; Elisa Cerdá-Doñate, Investigation, Writing - review and editing; Hubert M Taïeb, Software, Writing - review and editing; Tom Robinson, Damien Faivre, Stefan Klumpp, Conceptualization, Supervision, Writing - review and editing

**Author ORCIDs**
Hubert M Taïeb ⓘ http://orcid.org/0000-0002-7530-8988
Tom Robinson ⓘ http://orcid.org/0000-0001-5236-7179
Damien Faivre ⓘ http://orcid.org/0000-0001-6191-3389
Stefan Klumpp ⓘ http://orcid.org/0000-0003-0584-2146

**Decision letter and Author response**
Decision letter https://doi.org/10.7554/eLife.71527.sa1
Author response https://doi.org/10.7554/eLife.71527.sa2

## Additional files

### Supplementary files
• Transparent reporting form

### Data availability
All data (experimental and simulated trajectories) as well as analysis and simulation code has been deposited at Edmond, https://doi.org/10.17617/3.7b.

The following dataset was generated:

| Author(s) | Year | Dataset title | Dataset URL | Database and Identifier |
| --- | --- | --- | --- | --- |
| Charsooghi MA, Cerda-Donate E, Codutti A, Faivre D, Klumpp S, Robinson T | 2021 | Single-cell characterization of magnetotactic bacteria motion in high confinement | https://doi.org/10.17617/3.7b | Edmond, 10.17617/3.7b |

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
