## [Editor Report]

The manuscript reports results of a combined experimental and numerical investigation of magnetotactic bacteria in strong spatial confinement and under the influence of an external magnetic field. Single cells are trapped in micrometer-sized microfluidic chambers, and a variety of trajectories are found, which depend on the chamber size and the strength of the magnetic field. A detailed understanding of swimming in simple controlled confinement is essential to predict the behavior of motile microorganisms in the complex environments of their natural habitat.

---

## [Decision Letter]

**Decision letter after peer review:**

Thank you for submitting your article "Single-cell motion of magnetotactic bacteria in microfluidic confinement: interplay between surface interaction and magnetic torque" for consideration by *eLife*. Your article has been reviewed by 2 peer reviewers, and the evaluation has been overseen by a Reviewing Editor and Anna Akhmanova as the Senior Editor. The reviewers have opted to remain anonymous.

Essential revisions:

1) It would greatly benefit the manuscript to analyze separately the chambers containing an isolated cell from the chambers with 2 or more cells. As Figure 1c shows this is not a negligible situation.

2) Figure 2 shows quite a good agreement between theory and experiments, however, the simulation curves are consistently broader and with a peak slightly shifted to the right. Can the authors elaborate on this (minor) mismatch?

3) The simulations of the confined microalgae used a dumbbell shape to represent the shape asymmetry of body+flagella, as far as I understand, and not the bare cell body shape. Magnetospirillum gryphiswaldense are elongated, curved, and with two flagella.

The authors compare their results with flagella at either end of cell. Thus, they are far from spherical. Have the authors considered shapes other than spherical in their modeling?

4) Furthermore, these bacteria may reverse their motion. Does confinement influence the rate of reversal? and does it depend on the size of the compartment?

5) Experimental studies of microswimmers in confinement and near planar and curved surfaces are not quite as rare as indicated in the introduction. See, e.g.

A.P. Berke et al., PRL 101, 038102 (2008);

P. Denissenko et al., PNAS 109, 8007 (2012);

V. Kanstler et al., PNAS 110, 1187 (2013).

Also, several theoretical and numerical studies have been performed, see, e.g.

Y. Fily et al., Soft Matter 10, 5609 (2014);

A. Wysocki et al., Phys. Rev. E 91, 050302 (2015);

S.E. Spagnolie et al., Soft Matter 11 3396 (2015);

S. Rode et al., New J. Phys. 21, 013016 (2019).

6) In the introduction of the trap sizes, it would be good to mention the length of the bacterial body for comparison.

7) The authors find that the observed swimming behavior in the absence of a magnetic field is very similar to the one reported for *Chlamydomonas* algae in confinement. This is quite surprising, because both the shape (rod versus asymmetric dumbbell) and the hydrodynamic interactions (neutral versus puller) should be quite different.

Thus, a significantly different behavior would be expected. This should be discussed in more detail.

8) It seems that the bacteria would be better described by active Brownian rods that be spheres, compare

G. Li et al., PRL 103, 078101 (2009); J. Elgeti et al., EPL 85, 38002 (2009).

9) Is the orientation vector of the active Brownian spheres confined to be always parallel to the top and bottom walls?

10) The Conclusion is more a Summary, and should be denoted accordingly.

11) It would be preferable to integrate the Supplementary Information into the main body of the text, rather than having it as a standalone file. Please consider doing so, using where appropriate Figure Supplements and expanding the methods section at the end of the paper.

---

## [Author Response]

Essential revisions:1) It would greatly benefit the manuscript to analyze separately the chambers containing an isolated cell from the chambers with 2 or more cells. As Figure 1c shows this is not a negligible situation.

Thank you for this suggestion, which points to an interesting question, namely whether there are interactions between the bacteria. First of all, we want to emphasize that we did experiments in such a way to make sure we typically observed individual bacteria. Figure 1c shows that by varying the concentration of bacteria (measured via the optical density) we can control the number of bacteria in a trap. For all following experiments we chose bacterial concentrations low enough that we got traps containing single bacteria. Only for the two largest trap sizes (25 µm and 45 µm), this was not possible and we analyzed traps that contained up to 3 bacteria (for the 25 µm traps, we got a mixture of traps containing 1-3 bacteria, for the 45 µm traps, we always had at least 2 bacteria per trap). For the smaller traps, the rare cases with more than one bacterium in a trap were not included in the analysis. This procedure is now explained in more detail in the text referring to Figure 1.

To address the reviewer question and to check if there are indeed differences in motion in a trap occupied by a single or more bacteria, reflecting the interactions between the bacteria, we analyzed the trajectories for the trap size 25 µm in more detail as suggested and included the results as a new Figure 7. For this trap size, we obtained traps containing either 1, 2 or 3 bacteria. We first plotted individual trajectories in Figure 7 —figure supplement 1; each row shows the output of each independent experiment and the columns correspond to different magnetic field strengths (0, 50, and 500 T). These trajectories are qualitatively similar between the cases with 1, 2 or 3 bacteria in a trap. To quantify possible differences, we determined U-turn radii (Figure 7). Here, the variation between experiments for the same conditions are similar or bigger than those between the cases with different number of bacteria. We thus conclude that interaction effects, if they are present, are small compared to the effect of the observed heterogeneity of bacteria (in swimming velocity and magnetic moment).

2) Figure 2 shows quite a good agreement between theory and experiments, however, the simulation curves are consistently broader and with a peak slightly shifted to the right. Can the authors elaborate on this (minor) mismatch?

Indeed, the experimental peaks are slightly broader to the right of the maximum (towards the wall), while the simulations are sharper there. At the same time, the simulated distributions are a bit broader to the left. We attribute this minor mismatch to systematic uncertainties in the experiments that make the wall less sharp and well-defined than in the simulations: On the one hand, there are inaccuracies in tracking the bacteria near the wall due to wall brightness and camera shaking. On the other hand, the wall may not be entirely regular or not perfectly vertical, so that there are small variations in trap radius.

Nevertheless, the chi square fit gives overall a good match between the experimental and simulated distributions. We have added a comment addressing the minor discrepancy in the corresponding part of the text on p.7.

3) The simulations of the confined microalgae used a dumbbell shape to represent the shape asymmetry of body+flagella, as far as I understand, and not the bare cell body shape. Magnetospirillum gryphiswaldense are elongated, curved, and with two flagella.The authors compare their results with flagella at either end of cell. Thus, they are far from spherical. Have the authors considered shapes other than spherical in their modeling?

This point is closely related to points 7 and 8 below, so we answer these questions together.

In our model, the shape of the bacteria (different from that of the algae studied by Ostapenko et al.), which is most important for the interaction with the walls, is implicitly included in the interaction parameters, specifically the wall torque. To make this clear, we have explicitly included a discussion of the case of an elongated rod-like shape representing the elongated cell body of the bacteria and the flagella emanating from their poles (also addressed in reviewer comment 8). In the description of the model in the methods section, we now show how a wall torque of the type used here can be motivated from an elongated geometry and steric interactions with the wall.

However, we view our approach as phenomenological rather than as a microscopic description of the detailed interaction with the wall, which we expect to be a combination of steric and hydrodynamic interactions, both in general dependent on the shape. Since the shape seems not to matter too much as seen by the similarities between the cases of our bacteria and the algae studied by Ostapenko et al., we speculate that the dominant contributions are an initial alignment parallel to the wall immediately upon a collision and eventually a disturbance of that alignment, which could be provided simply by rotational diffusion, then active motion after a rotation will carry the swimmer out of the reach of the relatively short-ranged wall interactions.

We have extended the discussion of these questions in the Discussion section, where we now address the comparison of different swimmer types, the elongated geometry as well as our general picture of wall collisions.

4) Furthermore, these bacteria may reverse their motion. Does confinement influence the rate of reversal? and does it depend on the size of the compartment?

The bacteria do indeed occasionally reverse their direction of motion. Following the reviewers’ suggestion, we have analyzed the frequency of reversals and included this analysis as a new Figure 3. The rates are relatively low and the persistence length of the motion exceeds the size of the traps, i.e., typically, bacteria swim over distances that exceed the sizes of all traps before a reversal occurs. No systematic dependence of the rates on either trap size or magnetic field strength was found (Figure 3 – supplement 1). An analysis of the reversal rate as a function of the distance from the wall is shown in figure 3 and also shows no systematic dependence on the distance from the wall. Therefore, we conclude that reversals are not (or at least not strongly) influenced by the confinement.

5) Experimental studies of microswimmers in confinement and near planar and curved surfaces are not quite as rare as indicated in the introduction. See, e.g.A.P. Berke et al., PRL 101, 038102 (2008);P. Denissenko et al., PNAS 109, 8007 (2012);V. Kanstler et al., PNAS 110, 1187 (2013).Also, several theoretical and numerical studies have been performed, see, e.g.Y. Fily et al., Soft Matter 10, 5609 (2014);A. Wysocki et al., Phys. Rev. E 91, 050302 (2015);S.E. Spagnolie et al., Soft Matter 11 3396 (2015);S. Rode et al., New J. Phys. 21, 013016 (2019).

We thank the reviewers for the suggested literature, we have extended the discussion of previous work in the introduction.

6) In the introduction of the trap sizes, it would be good to mention the length of the bacterial body for comparison.

Good suggestion, we added this on p.4.

7) The authors find that the observed swimming behavior in the absence of a magnetic field is very similar to the one reported for Chlamydomonas algae in confinement. This is quite surprising, because both the shape (rod versus asymmetric dumbbell) and the hydrodynamic interactions (neutral versus puller) should be quite different.Thus, a significantly different behavior would be expected. This should be discussed in more detail.

In our view, this point is closely related to point 3 above as well as point 8 below. We have extended the discussion in the Concluding remarks section to address this point. Indeed, the behavior in confinement is dominated by interactions with walls (one could even interpret our results as showing that the magnetic field only matters in weaker confinement/larger traps, where U turn radii are smaller than the trap radius). These interactions in general depend on the shape of the swimmer as well as its type of hydrodynamics. Nevertheless, the general behavior seems rather insensitive to the details of the propulsion mechanisms, as also noted by the second reviewer. In our extended discussion, we hypothesize that some initial alignment parallel to the wall (possible due to steric interactions) and a mechanism to eventually escape the wall (possibly due to reorientation by rotational diffusion plus active swimming away) are sufficient to obtain the general picture of transient swimming along the wall.

8) It seems that the bacteria would be better described by active Brownian rods that be spheres, compareG. Li et al., PRL 103, 078101 (2009); J. Elgeti et al., EPL 85, 38002 (2009).

We agree with the reviewer that the shape is indeed roughly rod-like. We discuss this together with the response to comment 3 above and have extended the discussion of the impact of shape in the paper, see point 3 above.

9) Is the orientation vector of the active Brownian spheres confined to be always parallel to the top and bottom walls?

No, the orientation vector is free to move in any direction; we perform full 3D simulations, where the bacterium undergoes wall interactions both at the side wall and at the top and bottom walls. Therefore, there is no confinement of the orientation vector, but rather the orientation is determined by the wall interactions. However, because of the small height of the traps, the bacteria interact with the top and bottom walls very frequently, so their orientation vector will typically be aligned parallel to the top and bottom walls. We have added a comment in the “Computational modeling” section of methods.

10) The Conclusion is more a Summary, and should be denoted accordingly.

We have renamed the section “Concluding remarks” to indicate this. Note that the section has also been extended in the revision.

11) It would be preferable to integrate the Supplementary Information into the main body of the text, rather than having it as a standalone file. Please consider doing so, using where appropriate Figure Supplements and expanding the methods section at the end of the paper.

This has been done. The Methods section has been expanded, Tables were included, mostly into the Methods section, and figures were re-organized as proposed.